# Genome-Wide Analyses Reveal Genetic Convergence of Prolificacy between Goats and Sheep

**DOI:** 10.3390/genes12040480

**Published:** 2021-03-26

**Authors:** Lin Tao, Xiaoyun He, Yanting Jiang, Yufang Liu, Yina Ouyang, Yezhen Shen, Qionghua Hong, Mingxing Chu

**Affiliations:** 1Key Laboratory of Animal Genetics, Breeding and Reproduction of Ministry of Agriculture and Rural Affairs, Institute of Animal Science, Chinese Academy of Agricultural Sciences, Beijing 100193, China; 82101182357@caas.cn (L.T.); hexiaoyun@caas.cn (X.H.); yufangliu@hebeu.edu.cn (Y.L.); 2Yunnan Animal Science and Veterinary Institute, Kunming 650224, China; jiangyanting-2007@163.com (Y.J.); yinaouyang@163.com (Y.O.); 3College of Life Science and Food Engineering, Hebei University of Engineering, Handan 056038, China; 4Annoroad Gene Technology Co., Ltd., Beijing 100176, China; yezhenshen@genome.cn

**Keywords:** litter size, genetic convergence, genome, RNA-seq, sheep, goats

## Abstract

The litter size of domestic goats and sheep is an economically important trait that shows variation within breeds. Strenuous efforts have been made to understand the genetic mechanisms underlying prolificacy in goats and sheep. However, there has been a paucity of research on the genetic convergence of prolificacy between goats and sheep, which likely arose because of similar natural and artificial selection forces. Here, we performed comparative genomic and transcriptomic analyses to identify the genetic convergence of prolificacy between goats and sheep. By combining genomic and transcriptomic data for the first time, we identified this genetic convergence in (1) positively selected genes (*CHST11* and *SDCCAG8*), (2) differentially expressed genes (*SERPINA14*, *RSAD2*, and *PPIG* at follicular phase, and *IGF1*, *GPRIN3*, *LIPG*, *SLC7A11*, and *CHST15* at luteal phase), and (3) biological pathways (genomic level: osteoclast differentiation, ErbB signaling pathway, and relaxin signaling pathway; transcriptomic level: the regulation of viral genome replication at follicular phase, and protein kinase B signaling and antigen processing and presentation at luteal phase). These results indicated the potential physiological convergence and enhanced our understanding of the overlapping genetic makeup underlying litter size in goats and sheep.

## 1. Introduction

As ruminant livestock, sheep and goats are globally distributed and play important roles in providing meat, milk, and fur to humans. One primary objective of sheep and goat breeding programs is to increase the yield by phenotypic and genetic selection, and female fecundity is a basic and crucial key. Litter size (LS), the lambing number per parturition per ewe or the kidding number per parturition per doe, is an important component of goat and sheep productivity. Importantly and interestingly, variation in individual LS is common within breeds, such as 1–4 in Yunshang black goats, 1–5 Romanov sheep, and 1–6 in Finnsheep [1,2]. Breeds with exceptional prolificacy can provide genetic material to develop new lines/populations with ideal performance.

It is important to elucidate the genetic mechanisms of LS to efficiently utilize these prolific resources. Therefore, numerous studies have been separately performed to enhance our understanding of goat and sheep LS [3,4,5,6,7]. Compared with sheep, knowledge of hircine prolificacy is in its infancy. Thus, previous studies explored the association between hircine LS and homologous genetic markers first identified in sheep (such as *FecB* and *FecX*) [8,9], in which the genetic convergence in prolificacy between goats and sheep was expected in principle, at least at the single gene level. However, the approach using candidate genes/loci did not work well. Notwithstanding the convergent genomic signatures of domestication and high-altitude adaptation that have been explored in sheep and goats [10,11], thus far, there has been a paucity of research on the genetic convergence of prolificacy from a whole-genome perspective. We hypothesize that the genetic convergence of prolificacy occurred in goats and sheep, likely because they were exposed to similar natural and/or artificial selection forces.

Therefore, the objective of this study was to identify the genetic convergence of prolificacy between goats and sheep using whole-genome data by examining (1) positively selected genes (PSGs), (2) differentially expressed genes (DEGs), and (3) biological pathways. These results will enhance our knowledge of prolificacy and can be used to improve the molecular design breeding of both goats and sheep.

## 2. Materials and Methods

### 2.1. Animals, Phenotypes and Sample Collection

This study included two Yunshang black goat populations (Yunshang black goats used for DNA sequencing (YSD), *n* = 40; Yunshang black goats used ovarian mRNA sequencing (YSR), *n* = 20) and three sheep populations (Finnsheep, FIN, *n* = 19; Romanov, ROM, *n* = 19; and Small Tailed Han sheep, STH, *n* = 12); the YSD, FIN and ROM data were obtained from previous studies [1,2]. All animals were as unrelated as possible as determined by pedigree or farmers’ knowledge. On the basis of average LS, the five populations were classified into a low-yield group (LG, control) and a high-yield group (HG, case) to represent extreme individuals within each breed (Appendix A). For each population, a highly significant difference was observed in LS between the HG and LG (*p*-value < 0.01, Mann–Whitney *U* test, Table 1).

To detect PSGs for LS, 20 YSD (LS ≤ 1.5), 9 FIN (LS ≤ 2), and 9 ROM (LS ≤ 1.8) were labeled as controls, and 20 YSD (LS > 2.3), 10 FIN (LS > 3.28), and 10 ROM (LS ≥ 3) were labeled as cases. Likewise, to identify DEGs for LS, 10 YSR (LS ≤ 2) and 6 STH (LS = 1) were selected as controls, and 10 YSR (LS ≥ 3) and 6 STH (LS > 2.66) were selected as cases. For ovary collection, estrus synchronization with controlled internal drug release (CIDR, progesterone 300 mg) for 16 and 12 days was separately performed for YSR and STH. Finally, 10 YSR (5 cases and 5 controls) and 6 STH (3 cases and 3 controls) were slaughtered within 45–48 h of CIDR removal for ovary harvest at follicular phase (FP), and the remaining 10 YSR and 6 STH were slaughtered on days 11 and 9 for ovary harvest at luteal phase (LP).

### 2.2. Whole-Genome Single Nucleotide Polymorphism (SNP)

Genomic DNA was extracted from the jugular venous blood of YSD and ear marginal tissues of FIN and ROM following a standard phenol–chloroform method. To obtain genomic SNPs of YSD, whole-genome sequencing of a single library was performed using the Illumina NovaSeq 6000 system (Illumina Inc., San Diego, CA, USA), and 150-bp paired-end reads were generated. To obtain clean reads, the following unqualified raw reads were removed: (1) adaptor-polluted reads, (2) reads with over 50% bases with a Phred score <19, and (3) reads with over 5% N bases. Clean reads were mapped to the ASR1 reference genome using a Burrows–Wheeler Aligner v0.7.9a [12], with an average depth of ~20-fold coverage, and SNPs were subsequently called using a Genome Analysis Toolkit v3.3 [13]. FIN and ROM were genotyped by the Ovine Infinium HD BeadChip, which comprises 606,006 SNPs. To obtain eligible SNPs and samples, the following quality control criteria were implemented in Plink v1.90 [14]: (1) call rate >90%, (2) SNPs on autosomes, (3) minor allele frequency >1%, (4) missing genotype frequency <10%, and (5) Hardy–Weinberg disequilibrium *p*-value > 10^−6^.

### 2.3. Population Structure

To explore the genetic relationship of individuals from YSD, FIN and ROM, Plink command “--indep-pairwise 50 5 0.2” was run to generate independent effective SNP sets for principal component analysis [14].

### 2.4. Selective Sweeps

To detect PSGs for LS, the fixation index (Fst), which is a measure of population differentiation, was calculated in VCFtools v0.1.14 [15] using Weir and Cockeram’s method [16] to compare LG and HG for YSD (Fst(YSD)), FIN (Fst(FIN)) and ROM (Fst(ROM)). For YSD, a 100-kb window with a 50-kb step size was used. For sheep, Fst(FIN) and Fst(ROM) were estimated SNP by SNP, and their root mean square (Fst_RMS) was then calculated with the following formula:$$\mathrm{Fst}\_\mathrm{RMS}=\sqrt{\frac{1}{2}\;(\mathrm{Fst}{\left(\mathrm{FIN}\right)}^{2}+\mathrm{Fst}{\left(\mathrm{ROM}\right)}^{2})}$$

To calculate Fst_RMS, only SNPs shared by FIN and ROM and those with nonnegative Fst values were used. The results were visualized using R package qqman [17].

To identify loci with extremely high Fst values that drive reproductive differences of LG and HG, we ranked the regions/loci in descending order on the basis of Fst(YSD) and Fst_RMS, respectively. The regions with extremely high Fst(YSD) (0.05, top 0.8%) were treated as candidates for goats, and the SNPs with extremely high Fst_RMS (0.3, top 0.7%) were selected as candidates for sheep. Finally, PSGs wholly or partially located at these regions or covering candidate SNPs were detected by ANNOVAR v2020-06-07 [18].

To explore the association of three *HS3ST2* variations and reproductive performance, an additional 544 2–5-year-old female Yunshang black goats were genotyped using a MassARRAY^®^ SNP system. The primer sequences are in Appendix A. Chi-squared test (all expected frequencies ≥ 5) or Fisher’s exact test were employed to detect associations between the genotype and LS of Yunshang black goats.

### 2.5. Ovarian RNA-Seq

The total RNA of hircine ovaries for LG and HG at FP and LP (five biological replicates) was extracted and sequenced using the Illumina NovaSeq 6000 platform (Illumina Inc., San Diego, CA, USA). After filtering out the adapter reads and unqualified reads, clean reads were mapped to the ASR1 reference genome using HISAT2 [19], followed by the assembly and calculation of read count and fragments per kilobase per million values in StringTie [20], and identification of DEGs with DESeq2 [21]. In the two comparisons of LG vs. HG, genes were accepted as being differentially expressed when (1) *q*-value (*p*-value adjusted by the Benjamini–Hochberg method) <0.5, and |Log_2_FoldChang| ≥ 0.24. Likewise, the total RNA of 12 ovine ovaries for LG and HG at FP and LP (three biological replicates) were subjected to the same processing, except they were mapped to the Oar_v3.1 reference genome.

### 2.6. Identification of Genetic Convergence

To reveal the potential genetic convergence in goat and sheep prolificacy, we focused on (1) PSGs, (2) DEGs, and (3) biological pathways at the level of gene set. Given the availability of PSGs in HG using the number of segregating sites by length (nSL), a haplotype-based statistic with test power to both soft and hard sweeps [2,22], convergent PSGs were identified as the intersection of ovine candidates identified by Fst_RMS and hircine candidates shared by Fst(YSD) and nSL (≥6) of HG. Convergent DEGs at LP and FP were defined as these genes that were shared by goats and sheep, but excluded their expression trend. The number of genes shared by Fst(YSD) and nSL of HG is limited and nonideal for gene set analysis; therefore the Gene Ontology (GO) and Kyoto Encyclopedia of Genes and Genomes (KEGG) pathway analyses were performed for the PSGs on the basis of Fst(YSD) and Fst_RMS using Metascape [23] to obtain the related gene set which was composed of functionally associated PSGs. Terms with *p*-value < 0.01 were considered to be significantly different. Similarly, DEGs were also used to identify the connected gene set for prolificacy between goats and sheep. Therefore, convergent biological pathways were identified as the mutual GO (including only biological processes) and KEGG Ontology (ko) terms.

## 3. Results and Discussion

### 3.1. Population Structure

The principal component analysis (Appendix A) demonstrated a clear split between FIN and ROM along the first principal component (PC), indicating that they were independent populations. However, they were northern European short-tailed breeds with common ancestry components [24]. Therefore, they are thought to be useful for exploring the genetic mechanisms underlying LS as single and separate herds. For both goats and sheep, the cases and controls within each breed were not split by the first two PCs, which was consistent with the circumstances of dogs in a previous report on behavior [25]. These findings indicate that population structure was not linked to LS of each breed.

### 3.2. Convergent PSGs Linked to LS

To reveal the signatures of selection for goat prolificacy, a set of 350 PSGs was identified by Fst(YSD), of which *HS3ST2* on chromosome 25 was found to have the strongest selection signature (Figure 1 and Appendix A). The variation g.20423876C>T at *HS3ST2* was significantly associated with LS (Appendix A). Coding heparan sulfate-glucosamine 3-sulfotransferase 2, *HS3ST2* was reported to be linked to circadian rhythm regulation in the pineal gland of rats [26]. *HS3ST5*, an *HS3ST2* paralog, was associated with reproductive seasonality traits in sheep [27]. Given the association of seasonality-associated *DIO3* and LS in pigs and goats, and the empirical concurrence of higher LS and year-round estrus in goats [2,28], these results indicate that *HS3ST2* may be involved in female reproductive health, and seasonal breeding in goats.

Eight overlapping PSGs were revealed on the basis of Fst(YSD) and nSL of HG in goats: *CHST11*, *SDCCAG8*, *ATF6*, *FAM13A*, *HSP90B1*, *IBSP*, *IFNLR1*, and *STAB2* (Figure 2). Of these, two genes (*CHST11* and *SDCCAG8*) were also under selection in sheep for which 162 PSGs were identified by Fst_RMS (Figure 1 and Figure 2). This number was significantly larger than expected (*p*-value = 0.05), suggesting that *CHST11* and *SDCCAG8* are significantly convergent in goat and sheep prolificacy. *CHST11* (also known as *C4ST-1*), which encodes carbohydrate sulfotransferase 11, is a target gene of transforming growth factor β (TGFβ)-like growth factors (such as TGFβ, BMP2, and Activin) [29]. *Chst11* was found to regulate the proliferation and apoptosis of embryo chondrocytes in mice [30]. The mRNA level of *CHST11* increases upon the interaction of BMP4 with a phospho-Smad3 binding site in the CHST11 promoter [31]. Furthermore, *CHST11* was linked to ovarian cancer [32], endometrial carcinomas [33], and porcine farrowing interval [34]. However, the CHST11 transcripts were not differentially expressed in goat ovary (Appendix A). Another PSG, *SDCCAG8*, is a risk gene for Bardet–Biedl syndrome, which is primarily characterized by postaxial polydactyly, hypogonadotropic hypogonadism, and/or genitourinary malformations, and secondly characterized by polycystic ovary syndrome [35]. *SDCCAG8* is a candidate gene for hind limb and rib cage malformations in mice [36,37] and teat number in pigs [38]. Interestingly, *BMPR1B*, a major gene for sheep LS, plays important roles in both chondrogenesis and embryogenesis [39,40], indicating the pleiotropism of candidate genes for bone formation and reproduction. This was, to some extent, supported by the “osteoclast differentiation” pathway, which was enriched by some PSGs for LS (Table 2). Moreover, there is a transcription factor binding site at *SDCCAG8* for SMAD5, a member of the TGFβ signaling pathway, and strong differentiation signals have been observed in SMAD5 of Shaanbei White Cashmere goats with different LS [7]; this may contribute to the differential expression of *SDCCAG8* at both LP and FP (Appendix A). Taken together, these findings indicate that *CHST11* and *SDCCAG8* regulate LS via TGFβ and BMP signaling in goats and sheep.

### 3.3. Convergent DEGs Associated with LS

For goats, compared with LG, 2207 (1126 up- and 1081 down-regulated) and 1665 DEGs (826 up- and 839 down-regulated) were identified in HG at LP and FP, respectively (Figure 3 and Appendix A). For sheep, 104 and 39 DEGs were identified in HG compared with LG at LP and FP, respectively (Figure 3 and Appendix A). In total, eight overlapping DEGs were shared by goats and sheep, including three (*SERPINA14*, *RSAD2*, and *PPIG*) at FP and five (*IGF1*, *GPRIN3*, *LIPG*, *SLC7A11*, and *CHST15*) at LP (Figure 2 and Figure 4). Of these, *IGF1* and *SLC7A11* were identified as DEGs in the transcriptomic analysis of ovaries from pigs with high and low LS [41]. Additionally, *SLC7A11* is differentially expressed in ovaries among sheep breeds with different fecundity [42]. *IGF1* is a candidate gene for LS in goats and sheep [43,44,45], suggesting the convergence at the levels of DNA and change in mRNA expression.

Previous studies reported the differential expression of LIPG mRNA or protein in the theca interna during the transition from small to large antral bovine follicles [46], in the placenta of Berkshire pigs with different LS [47], and in the human follicular fluid of poor ovarian responders compared with a control group [48]. Additionally, a genome-wide association study on Large White pigs identified *LIPG* as a candidate gene for piglet uniformity measured as the coefficient of variation of birth weights within one litter [49]. These findings highlight the potential roles in follicular development and pregnancy. Secreted from uterine endometrium and induced by progesterone, uterine serpin (SERPINA14) plays crucial roles in the pregnancy of farm animals, such as nutrient supply to the conceptus, immunoregulation, and maternal recognition of pregnancy [50,51,52]. *RSAD2* is a marker for dendritic cell maturation [53] and early pregnancy diagnosis in dairy cows [54]. Moreover, compared with healthy follicles, *RSAD2* was up-regulated in early atresia and progressive atresia groups in pigs [55]. The functions of *GPRIN3*, *CHST15*, and *PPIG* in reproduction are unclear. Collectively, it is likely that these DEGs influence female reproduction by multiple processes, such as follicular development, immunoregulation, and pregnancy maintenance.

### 3.4. Convergent Biological Pathways

At the single gene level, it is possible to infer convergent signals from overlapping DEGs or PSGs between goats and sheep. At the gene set level, GO and KEGG pathway analyses for PSGs and DEGs were implemented to identify the overlapping biological pathways that include a series of functionally connected genes. For goat PSGs, 192 GO and 30 ko terms were revealed, as expected, six acrosome reaction-related GO terms were detected: binding of sperm to zona pellucida (GO:0007339), regulation of acrosome reaction (GO:0060046), single fertilization (GO:0007338), regulation of fertilization (GO:0080154), sperm–egg recognition (GO:0035036), and fertilization (GO:0009566) (Appendix A). Additionally, some endocrine resistance ko terms were identified: endocrine resistance (ko01522), estrogen signaling pathway (ko04915 and hsa04915), relaxin signaling pathway (hsa04926), and fluid shear stress and atherosclerosis (hsa05418) (Appendix A). For sheep PSGs, we identified 132 GO and 58 ko terms, of which osteoclast differentiation (ko04380 and hsa04380), ErbB signaling pathway (hsa04012 and ko04012), and relaxin signaling pathway (hsa04926) were shared by goats (Figure 2 and Table 3).

For goat DEGs, as expected, the phosphatidylinositol 3-kinase (PI3K)-Akt signaling pathway (ko04151), oocyte meiosis (ko04114), progesterone-mediated oocyte maturation (ko04914), and estrogen signaling pathway (ko04915) were in the top 10 ko terms (Appendix A), which reemphasizes the significance of some biological processes in LS. Consequently, goats shared three terms with sheep: protein kinase B signaling (GO:0043491) and antigen processing and presentation (ko04612) at LP, and the regulation of viral genome replication (GO:0045069) at FP (Table 3 and Figure 2). An implication of these convergent biological pathways is that there may be differences in relaxin concentration during pregnancy, immunity, and disease resistance that contribute to LS in goats and sheep. Moreover, it is helpful to pinpoint the potential causal genes for LS, such as *ADCY3*, which is necessary for male fertility [56]; *TGFB2*, which participates in ovine fecundity [57]; and *IGFBP5*, which is associated with LS in pigs [58]. Therefore, the convergence of biological pathways from the perspective of gene sets and single genes, indicates potential physiological convergence and enhances our understanding of the genetic basis underlying goat and sheep LS.

From a genomic and transcriptomic standpoint, we attempted to provide a genome-wide landscape to reveal the genetic convergence of prolificacy between goats and sheep. The results may indicate the physiological convergence of goats and sheep; however, more research is needed to characterize reproductive hormones and biological factors. However, these results may differ among other populations/breeds because of the sample size and/or genetic backgrounds in this study. Therefore, it would be useful to integrate an abundance of multi-omics data, such as genomics, transcriptomics, proteomics, and metabolomics, for additional populations or reproductive organs in a large population.

## 4. Conclusions

By combining genomic and transcriptomic data, this study identified the genetic convergence of prolificacy between goats and sheep for the first time on the basis of (1) PSGs (*CHST11* and *SDCCAG8*), (2) DEGs (*SERPINA14*, *RSAD2*, and *PPIG* at FP, and *IGF1*, *GPRIN3*, *LIPG*, *SLC7A11*, and *CHST15* at LP), and (3) biological pathways (PSG: osteoclast differentiation, ErbB signaling pathway and relaxin signaling pathway; DEG: regulation of viral genome replication at FP, and protein kinase B signaling and antigen processing and presentation at LP). These results indicated potential physiological convergence and enhanced our understanding of the overlapping genetic makeup underlying LS between goats and sheep.

## Figures and Tables

**Figure 1 genes-12-00480-f001:**
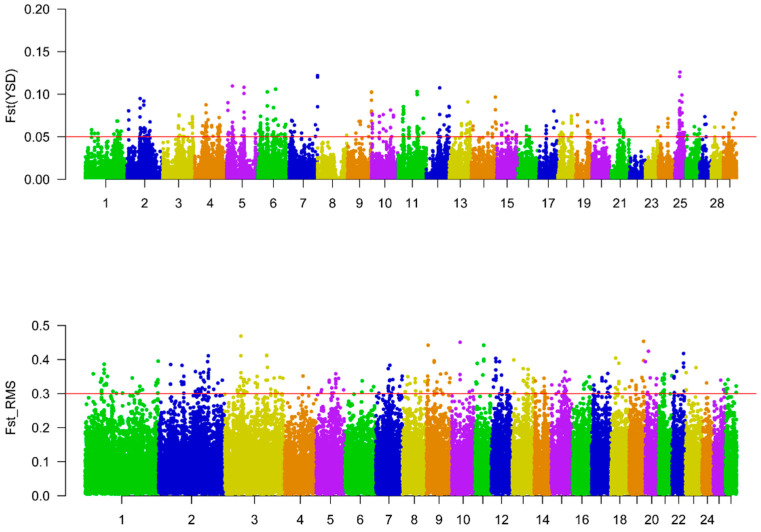
Manhattan plots for fixation index (Fst)(YSD) and root mean square of fixation index (Fst_RMS). The red lines indicate the cut-offs of Fst(YSD) (0.05, top 0.8%) and Fst_RMS (0.3, top 0.7%).

**Figure 2 genes-12-00480-f002:**
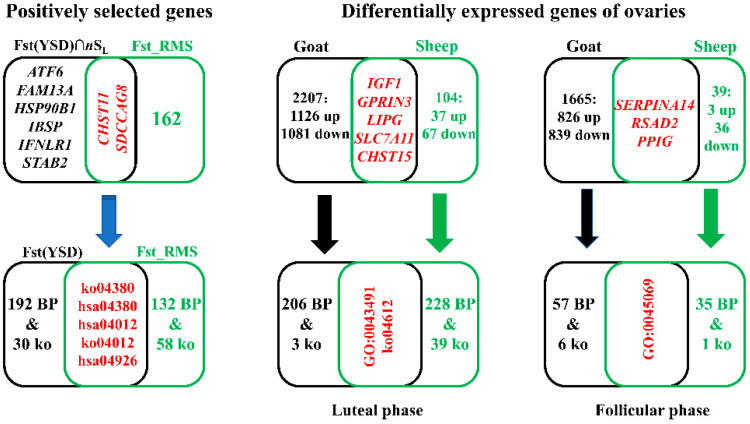
Venn diagrams of genetic convergence of prolificacy between goats and sheep. The top panels denote the convergence at the single gene level (positively selected genes and differentially expressed genes of ovaries), whereas the convergence at the gene set level is represented as Gene Ontology (GO) and Kyoto Encyclopedia of Genes and Genomes (KEGG) Ontology (ko) terms in the bottom panels; only biological processes (BP) were considered in GO analysis.

**Figure 3 genes-12-00480-f003:**
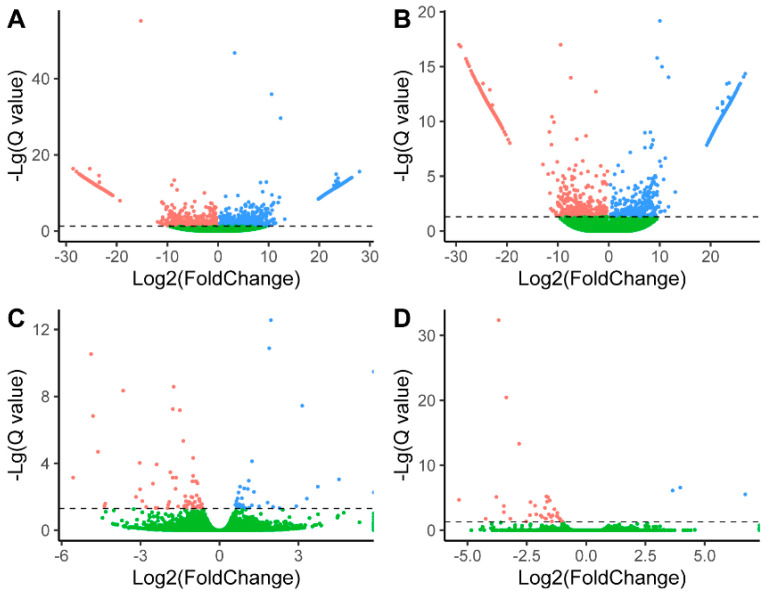
Volcano plots of differentially expressed genes in the ovaries of Yunshang black goats (**A**,**B**) and Small Tail Han sheep (**C**,**D**). (**A**,**C**) indicate luteal phases, and (**B**,**D**) denote follicular phases. Compared with the low-yield groups, points in red, blue and green are the down-regulated, up-regulated, and non-differentially expressed genes in high-yield groups, respectively. The dashed lines are thresholds of 1.30 (−lg(0.05)).

**Figure 4 genes-12-00480-f004:**
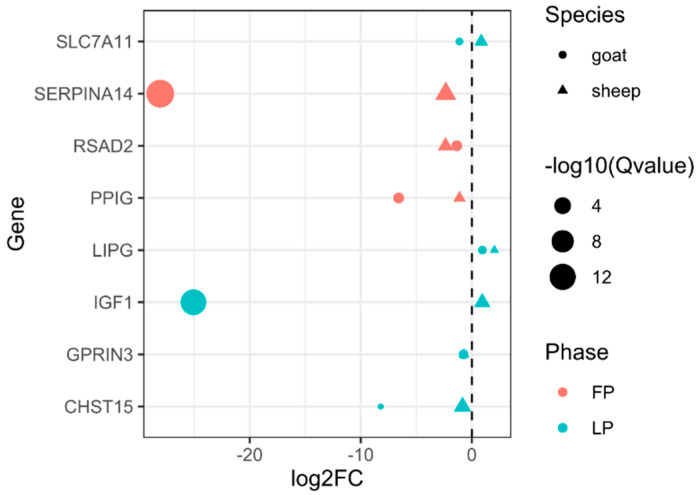
Differentially expressed genes of ovaries shared by sheep and goats. FP and LP denote follicular and luteal phases, respectively.

**Table 1 genes-12-00480-t001:** Phenotypes of does and ewes used in this study.

Population ^1^	Average Litter Size (Mean ± SD)		*p*-Value ^2^
	Low-Yield Group	High-Yield Group	
YSD	1.32 ± 0.19 (*n* = 20)	3.00 ± 0.38 (*n* = 20)	5.16 × 10^−8^
YSR	1.75 ± 0.26 (*n* = 10)	3.30 ± 0.35 (*n* = 10)	1.21 × 10^−4^
FIN	1.66 ± 0.33 (*n* = 9)	3.59 ± 0.41 (*n* = 10)	2.76 × 10^−4^
ROM	1.53 ± 0.25 (*n* = 9)	3.13 ± 0.17 (*n* = 10)	2.01 × 10^−4^
STH	1.00 ± 0.00 (*n* = 6)	2.89 ± 0.17 (*n* = 6)	2.22 × 10^−3^

^1^ YSD and YSR indicate Yunshang black goats used for DNA sequencing and ovarian mRNA sequencing, respectively. FIN: Finnsheep; ROM: Romanov sheep; STH: Small Tailed Han sheep. ^2^ Mann–Whitney *U* test. The number in the parentheses is sample size.

**Table 2 genes-12-00480-t002:** Pathways shared by positively selected genes of sheep and goats.

Term	Name	−lg(*p*)		Gene	
		Sheep	Goat	Sheep	Goat
hsa04012	ErbB signaling pathway	3.92	2.18	*CBL PIK3CA MAPK10 KCNH8 NRG4*	*CAMK2B GRB2 MAPK10 SRC AKT3*
ko04380	Osteoclast differentiation	3.27	2.89	*MAPK14 NFATC2 PIK3CA MAPK10 TGFB2*	*FCGR2A FCGR2B FCGR3A FCGR3B GRB2 MAPK10 TEC AKT3*
hsa04380	Osteoclast differentiation	3.14	2.72	*MAPK14 NFATC2 PIK3CA MAPK10 TGFB2*	*FCGR2A FCGR2B FCGR3A FCGR3B GRB2 MAPK10 TEC AKT3 RHBDF2*
ko04012	ErbB signaling pathway	3.01	2.34	*CBL PIK3CA MAPK10 NRG4*	*CAMK2B GRB2 MAPK10 SRC AKT3*
hsa04926	Relaxin signaling pathway	2.29	2.13	*MAPK14 PIK3CA MAPK10 CREB5*	*ADCY3 GNG5 GRB2 MMP9 MAPK10 SRC AKT3*

**Table 3 genes-12-00480-t003:** Terms shared by differentially expressed genes of sheep and goats.

Term	Name	−lg(*p*)		Gene	
		Goat	Sheep	Goat	Sheep
Protein kinase B signaling	GO:0043491	2.99	2.58	*TCF7L2 IGFBP5 RPS6KB2 MYOC LOC102175889*	*IGF1 KIT LOX TRK2 PKHD1*
Antigen processing and presentation	ko04612	2.08	3.86	*PDIA3 CREB1 CALR NFYC*	*HLA-A HLA-B* *LA-C HLA-F*
Regulation of viral genome replication	GO:0045069	2.25	6.15	*RSAD2 VAPA*	*IFIT1 MX1 ISG15 RSAD2*

## Data Availability

The raw sequencing data of 40 Yunshang black goats (YSD) have been deposited in the NCBI SRA database under accession code PRJNA611688. Genotype and phenotype datasets for Finnsheep and Romanov sheep were available at the following link: https://www.animalgenome.org/repository/pub/CAAS2018.0302/ (accessed on 24 September 2020). The RNA-seq data reported in this article are available upon request for research purpose.

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
