# Peer review of "Genome-Wide Analyses Reveal Genetic Convergence of Prolificacy between Goats and Sheep"

_genes, 2021, doi:10.3390/genes12040480_

Round 1
Reviewer 1 Report
The manuscript, “Genome-wide analyses reveal genetic convergence of prolificacy between goats and sheep” uses a number of different methods to identify variants and genes associated with litter size and expression in the ovaries. It was interesting to see the prolificacy of these two species together.
Dividing each breed into low yield and high yield to study litter size doesn’t seem like it could be useful. Prolificacy is controlled by many genes and is a quantitative trait, so dividing into two categories doesn’t fit the phenotype well. Additionally, within the low yield and high yield groups animals with very different litter sizes are grouped together. For example, a Finnsheep that had a litter size of 3 was included in the low yield group, while the maximum litter size in the Small tailed Han sheep was 3 (and those sheep with a litter size of 3 were included in the high yield group). While I realise this is the best design for the data presented here, please explain further for the reader about how low yield and high yield groups were determined and the justification in designing the study this way.
The extensive use of acronyms made this paper extremely challenging to read. With breeds, traits, replicates, and methods all identified with acronyms, the manuscript, at times, is very difficult to follow. In addition:
-“PSGs” was defined in this paper as both “signatures of selection at DNA level” (Line 133) and “positively selected genes” (Line 69). I realise you are referring to the same thing, but, keep it simple, and just use “positively selected genes”.
-“DEGs” was defined in this paper as both “changes in gene expression at RNA level” (Line 134) and “differentially expressed genes” (Line 71). As with “PSGs”, just keep it simple and use “differentially expressed genes”.
-Define “ko” (I believe it was the acronym for KEGG ontologies, but was never defined in the paper).
Finally, I would like to see your results discussed from the analyses of the association between three variants in HS3ST2 and reproductive performance in Yunshang black goats, Jining Gray goats and Liaoning cashmere goats. I believe looking at the association of single DNA markers with a trait using one breed as a high yield group and a different breed as a low yield in a test is complicated by the population structure, so it would be nice to read about the justification of doing the analyses this way.
Author Response
Reviewer 1#
The manuscript, “Genome-wide analyses reveal genetic convergence of prolificacy between goats and sheep” uses a number of different methods to identify variants and genes associated with litter size and expression in the ovaries. It was interesting to see the prolificacy of these two species together.
Response: Thanks for your useful comments and kind suggestions which are helpful to improve our manuscript. Please see our responses and revised version.
Dividing each breed into low yield and high yield to study litter size doesn’t seem like it could be useful. Prolificacy is controlled by many genes and is a quantitative trait, so dividing into two categories doesn’t fit the phenotype well. Additionally, within the low yield and high yield groups animals with very different litter sizes are grouped together. For example, a Finnsheep that had a litter size of 3 was included in the low yield group, while the maximum litter size in the Small tailed Han sheep was 3 (and those sheep with a litter size of 3 were included in the high yield group). While I realise this is the best design for the data presented here, please explain further for the reader about how low yield and high yield groups were determined and the justification in designing the study this way.
Response: Thanks for your insightful comments. For litter size of goats and sheep, we politely tend to acknowledge that it is a threshold trait controlled by polygenes. Here, we focused on average litter size but not single record to evaluate the female reproductive performance within each breed. Thus, it may not be helpful to compare litter size of a specific parity across breeds.
For YSD, FIN and ROM used in this study, the values of litter size were distributed in the two tails in a population of 277, 54 and 78 individuals, respectively. Based on individual average litter size (LS) from at least two parities (except one ROM sheep), 20 YSD (LS ≤ 1.5), 10 YSR (LS ≤ 2), six STH (LS = 1), 9 FIN (LS ≤ 2) and 9 ROM (LS ≤ 1.8) were labeled as low-yield groups, and 20 YSD (LS > 2.3), 10 YSR (LS ≥ 3), six STH (LS > 2.66),10 FIN (LS > 3.28) and 10 ROM (LS ≥ 3) were labeled as high-yield groups, respectively. Relatively speaking, we believe that these animals can represent the extreme ones within each breed. We also thank you for your understanding on this design.
Thanks again!
The extensive use of acronyms made this paper extremely challenging to read. With breeds, traits, replicates, and methods all identified with acronyms, the manuscript, at times, is very difficult to follow. In addition:
Response: Thank you very much. We now add the section of abbreviations (page 10) in the revised manuscript.
-“PSGs” was defined in this paper as both “signatures of selection at DNA level” (Line 133) and “positively selected genes” (Line 69). I realise you are referring to the same thing, but, keep it simple, and just use “positively selected genes”.
Response: Thanks. We use positively selected genes (PSGs) in the revised manuscript.
-“DEGs” was defined in this paper as both “changes in gene expression at RNA level” (Line 134) and “differentially expressed genes” (Line 71). As with “PSGs”, just keep it simple and use “differentially expressed genes”.
Response: Corrected as suggested.
-Define “ko” (I believe it was the acronym for KEGG ontologies, but was never defined in the paper).
Response: Thanks. We define it now.
Finally, I would like to see your results discussed from the analyses of the association between three variants in HS3ST2 and reproductive performance in Yunshang black goats, Jining Gray goats and Liaoning cashmere goats. I believe looking at the association of single DNA markers with a trait using one breed as a high yield group and a different breed as a low yield in a test is complicated by the population structure, so it would be nice to read about the justification of doing the analyses this way.
Response: Thanks for your suggestion. The results of association between three variations at HS3ST2 and reproductive performance were shown in Table S4. The litter size of Yunshang black goats was collected, whereas it was unavailable for Jining Gray and Liaoning cashmere goats (lines 132-133). Therefore, we used two tests to explore the association between markers and reproductive performance, including one testing genotype and litter size of Yunshang black goats, and another testing genotype and breed (the low-yield Liaoning cashmere goats vs high-yield Jining Gray goats).

Reviewer 2 Report
This manuscript describes multiple approaches to identify shared genes/pathways that contribute to increased litter size in sheep & goats. The authors utilize SNP chip data, whole genome sequencing, and RNA expression data to pursue their hypotheses. They identify several shared genes and pathways through these approaches providing new insight into shared genetic contributions to litter size in small ruminants. However there are certain aspects of the manuscript that need to be improved prior to acceptance. Specifically clarifying areas which need more detail and improving the English to ensure the reader can understand the authors meanings.
L33: “meant” to “meat”
L34: “phenotypical” to “phenotypic”
L41-42: This sentence needs to be reworked to be clearer in meaning.
L48: remove “,even in vain”
L54: “is” to “was” – past tense is recommended when describing scientific studies that were completed.
- Materials & Methods
How old were the animals when litter size was recorded? Was there a minimum age for inclusion? I’m concerned that there are differences between the low and high yield groups in the number of opportunities to lamb/kid. Specifically, in the YSD and FIN populations which have 2.5 vs 3.3 and 4.4 vs 5.3 kidding/lambing events per the low and high yield groups respectively. This discrepancy could inadvertently lead to identifying variants associated with longevity or something other than just litter size.
L64: unrelated within 1 generation? 2 or 5? Larger? Please be more specific.
L87: What size were the reads? Paired-end or single? Approximately what was the coverage generated from whole-genome sequencing.
L88: “Clear” reads meaning what? Those that passed quality control? If so, what metrics were used?
L92: It may help the reader to briefly list what the QC parameters were rather than cite a study and make them search for it.
L94: “form” to “from”
L116: “premier” to “primer”
L130: Why was Oar_v3.1 used when a newer and better built annotated assembly is available: rambouillet_v1.0?
Figure 2 appeared unclear in my document. Perhaps a higher dpi figure is needed to make the data presented there easier to read.
L175: “selected” to “selection
L181-183: This statement may need a citation.
L245: “DGEs” to “DEGs”
L245-246: I disagree that the approach would be difficult. It may take time and some computational resources but that shouldn’t prevent that authors from pursing this approach.
L249-250: Why were the acrosome related GO terms expected? Perhaps the authors could expand if they have evidence that the differences in litter size are due to failed fertilization, early embryonic loss, or diverging ovulation rates.
L275-281: Why were the other breeds (Icelandic, Hu, etc) present in Xu et al 2018 not utilized in the present study? The authors note more data from other populations and breeds may provide more evidence to support their conclusions yet they appear to have chose not to use data available to them.
Author Response
Reviewer 2#
This manuscript describes multiple approaches to identify shared genes/pathways that contribute to increased litter size in sheep & goats. The authors utilize SNP chip data, whole genome sequencing, and RNA expression data to pursue their hypotheses. They identify several shared genes and pathways through these approaches providing new insight into shared genetic contributions to litter size in small ruminants. However there are certain aspects of the manuscript that need to be improved prior to acceptance. Specifically clarifying areas which need more detail and improving the English to ensure the reader can understand the authors meanings.
Response: Thank you for your time spent on reviewing our manuscript. We sincerely appreciate your valuable comments which have definitely helped us to improve our manuscript. We have improved our English with the help of one native speaker. Please see our revised manuscript and our responses to your comments in the following.
L33: “meant” to “meat”
Response: Thanks. Revised.
L34: “phenotypical” to “phenotypic”
Response: Thanks. Revised.
L41-42: This sentence needs to be reworked to be clearer in meaning.
Response: Thank you. We rephrase it now.
It is important to elucidate the genetic mechanisms of LS to efficiently utilize these prolific resources.
L48: remove “, even in vain”
Response: Thanks. Do as suggested.
L54: “is” to “was” – past tense is recommended when describing scientific studies that were completed.
Response: Thanks. We corrected it according to your suggestion.
- Materials & Methods
How old were the animals when litter size was recorded? Was there a minimum age for inclusion? I’m concerned that there are differences between the low and high yield groups in the number of opportunities to lamb/kid. Specifically, in the YSD and FIN populations which have 2.5 vs 3.3 and 4.4 vs 5.3 kidding/lambing events per the low and high yield groups respectively. This discrepancy could inadvertently lead to identifying variants associated with longevity or something other than just litter size.
Response: Thanks for your insightful comments.
Generally, the animals gave birth for the first at the age of ~1.5 years old when litter size was recorded. Importantly, the phenotype involved in this study is not equivalent to their lifetime reproductive performance. Especially, most individuals of both low- and high-yield groups gave birth 2 to 3 parities in YSD population, which is lower than normal reproductive levels. What’s more, the parity number between low- and high-yield groups is not statistically significant in FIN population (p-value = 0.3, Mann-Whitney U test).
By the way, inspired by your concerns, it will be interesting to explore the phenotypic and genetic relationships between litter size and longevity in the future.
L64: unrelated within 1 generation? 2 or 5? Larger? Please be more specific.
Response: According to Xu et al., (2018), Finnsheep and Romanov were as unrelated as possible based on analysis of pedigree records and farmers’ knowledge. However, the pedigree is unavailable, making it difficult to determine the generation. We describe it more clearly.
Reference cited:
Xu, S.S.; Gao, L.; Xie, X.L.; Ren, Y.L.; Shen, Z.Q.; Wang, F.; Shen, M.; Eyϸórsdóttir, E.; Hallsson, J.H.; Kiseleva, T., et al. Genome-wide association analyses highlight the potential for different genetic mechanisms for litter size among sheep breeds. Front Genet 2018, 9,118.
L87: What size were the reads? Paired-end or single? Approximately what was the coverage generated from whole-genome sequencing.
Response: Paired-end reads with the length of 150 bp were generated in resequencing with an average depth of ~20-fold coverage.
L88: “Clear” reads meaning what? Those that passed quality control? If so, what metrics were used?
Response: We are sorry for the mistake. We meant the clean reads. To obtain clean reads, the following unqualified raw reads were removed: (1) adap-tor-polluted reads, (2) reads with over 50% bases with a Phred score < 19, and (3) reads with over 5% N bases.
L92: It may help the reader to briefly list what the QC parameters were rather than cite a study and make them search for it.
Response: Thank you. We changed it as suggested. Please see the revised manuscript for details.
L94: “form” to “from”
Response: Thanks. Changed.
L116: “premier” to “primer”
Response: Thanks. Changed.
L130: Why was Oar_v3.1 used when a newer and better built annotated assembly is available: rambouillet_v1.0?
Response: Thanks. Because the RNA-seq was performed before the release of rambouillet_v1.0. This point may not affect our main results and conclusions.
Figure 2 appeared unclear in my document. Perhaps a higher dpi figure is needed to make the data presented there easier to read.
Response: Thank you. We provide a 768 dpi figure now.
L175: “selected” to “selection
Response: Thanks. Corrected.
L181-183: This statement may need a citation.
Response: Thank you. We added one citation.
Bhattacharyya, S.; Feferman, L.; Tobacman, J.K. Regulation of chondroitin-4-sulfotransferase (CHST11) expression by opposing effects of arylsulfatase B on BMP4 and Wnt9A. Biochim Biophys Acta 2015, 1849, 342-352, doi:10.1016/j.bbagrm.2014.12.009.
L245: “DGEs” to “DEGs”
Response: Thanks. Changed.
L245-246: I disagree that the approach would be difficult. It may take time and some computational resources but that shouldn’t prevent that authors from pursing this approach.
Response: Thank you for suggesting this. We delete this sentence.
L249-250: Why were the acrosome related GO terms expected? Perhaps the authors could expand if they have evidence that the differences in litter size are due to failed fertilization, early embryonic loss, or diverging ovulation rates.
Response: Because litter size is the outcome of a series of physiological processes including the development of oocytes, fertilization, embryo implantation and development, where acrosome reaction is a prerequisite to successful fertilization / reproduction. To be honest, we also expect the evidences on ovulation and embryonic loss. Indeed, two terms including oocyte meiosis (ko04114) and progesterone-mediated oocyte maturation (ko04914) were found for goat DEGs (lines 290-292).
L275-281: Why were the other breeds (Icelandic, Hu, etc) present in Xu et al 2018 not utilized in the present study? The authors note more data from other populations and breeds may provide more evidence to support their conclusions yet they appear to have chose not to use data available to them.
Response: Good comments.
We agree that it is more convincing to combine the data from more populations. However, only Finnsheep and Romanov were included in this study because (1) we noted that the parity effect on litter size was not significant in Finnsheep and Romanov, but it is significant in other breeds (Wadi, Hu, Icelandic and Texel) (https://www.animalgenome.org/repository/pub/CAAS2018.0302/ Table 1); and (2) compared to Finnsheep and Romanov, we were not satisfied with the number of parity for Hu and Wadi, and the differences of individual average litter size between the top and bottom ~10 ewes for Icelandic and Texel. Therefore, we excluded the other four populations though their data were available.

Round 2
Reviewer 1 Report
The authors have mostly addressed comments to my satisfaction.
However, I don’t find the study design to detect the differences between low (Liaoning cashmere goats) or high (Jining Gray goats) yield goats to be justified. This approach only uses single markers and does not account for differences in the genetic background of both breeds and should not be included in this manuscript unless the authors also address the weaknesses in this approach for the reader.
Author Response
Reviewer 1#
The authors have mostly addressed comments to my satisfaction.
However, I don’t find the study design to detect the differences between low (Liaoning cashmere goats) or high (Jining Gray goats) yield goats to be justified. This approach only uses single markers and does not account for differences in the genetic background of both breeds and should not be included in this manuscript unless the authors also address the weaknesses in this approach for the reader.
Response: Thanks. We agree with you and remove this part.
Reviewer 2 Report
Thank you for making the recommended changes.
Author Response
Reviewer 2#
Thank you for making the recommended changes.
Response: Thank you.